# Co-construct, implement and evaluate a multi-level intervention to prevent a sedentary lifestyle in children—Study protocol of the CIPRES study

**Marion Carayol[1]\*, Steven Laujac[1,2]\*, Marie Cholley-Gomez[3], Josselin Franceschi[4], Axel Rozand[4], Laurence Pallier[4], Jean-Pierre Estrella[5], Bruno Vanhierde[5], Emma Guillet-Descas[6], Emmanuel Damville[7], Olivier Gavarry[1], Cyrille Delpierre[8]**

**1** IAPS Laboratory "Impact of Physical Activity on Health", University of Toulon, Toulon, France, **2** Centre Hospitalier Intercommunal Toulon-La Seyne sur Mer, Hôpital Sainte Musse, Toulon, France, **3** Laboratory Epsylon EA4556, Université Paul Valéry Montpellier & Epidaure - Institut du Cancer, Montpellier, France, **4** Comity of Health Education of Var Department (CODES 83), Toulon-La Garde, France, **5** Directorate of Departmental Services of National Education of Var (DSDEN 83), Academy of Nice, France, **6** Laboratory on Vulnerabilities and Innovation in Sport, Université Claude Bernard Lyon1, Villeurbanne, France, **7** Centre Hospitalier Spécialisé Pierrefeu du Var, Hôpital Henri Guérin, Pierrefeu-du-Var, France, **8** EQUITY Team, CERPOP UMR 1295, Inserm-Université Toulouse III, Toulouse, France

\* marion.carayol@univ-tln.fr (MC); laujacsteven@gmail.com (SL)

**Data Availability Statement:** No datasets were generated or analysed during the current study. All relevant data from this study will be made available upon study completion.

## Abstract

### Background

A sedentary lifestyle is commonly associated with a higher risk of chronic disease development. Among school-aged children from European countries, screen-time represents a significant portion of sedentary time with 39.8% of children spending more than 2h/day in front of a screen on average. Therefore, effective solutions to reduce sedentary behavior (SB) must be found. Multilevel interventions based on the socio-ecological model (SEM) are particularly relevant to take into account influences of the social environment on individuals' SB. Moreover, the trans-contextual model (TCM) can offer complementary levers for individuals' behavior change. The CIPRES study is a theory-based multilevel intervention designed to decrease the SB in French primary school children aged 8–10 years. The present paper describes the protocol of a randomized controlled study to evaluate the effectiveness of the CIPRES multilevel intervention on SB.

### Methods

The CIPRES study is a cluster-randomized controlled trial comparing intervention vs control groups. A total of 700 children are targeted for inclusion, distributed in four municipalities considered as clusters. The study consists of two successive phases: 1) co-building of a SB prevention intervention by using a participatory approach involving representatives of each level of the SEM (e.g., children, parents, staff from municipalities, teachers) and 2) implementation and evaluation of the intervention. The intervention will last for 6 weeks in each involved class. Primary outcome will be the sedentary time of children per week, assessed

**Funding:** This work was supported by the French Institut National du Cancer (INCa project grant PREV19-021, INCA_14185). https://www.e-cancer.fr/ SL received funding from the French Ministry of Health (ReSP-Ir 2021/N° enregistrement DGOS: RESPI-21-029). https://sante.gouv.fr/ The sponsors had no role in the study design, data collection and analysis, decision to publish, or preparation of the manuscript.

**Competing interests:** The authors have declared that no competing interests exist.

by accelerometry. In addition, children and their parents will be asked to fill out questionnaires concerning children's physical activity level, screen time, quality-of-life and variables of the TCM.

## Discussion

This study will give information on the effectiveness of a theory-based intervention, involving multiple levels of actors in the co-construction and the implementation of the intervention, that may interest schools and public health officers looking for innovative sedentary prevention programs.

## Introduction

### Sedentary: A public health issue

A less active or sedentary lifestyle is associated with higher mortality and the development of chronic pathologies including cancers (in particular colon, breast and endometrial cancer), type 2 diabetes, obesity, arterial hypertension, ischemic heart disease, osteoporosis [1–4]. Various studies indicate that the morbid processes associated with these pathologies and favored by sedentary behavior (SB) beginning in childhood or adolescence [5]. This period is conductive for the habit set up of the future adult. In particular, habits relating to physical activity (PA) practice, underlining the need to decrease the adoption of a sedentary lifestyle from an early age. According to Sedentary Behavior Research Network, SB is defined as any waking behavior characterized by an energy expenditure $\leq 1.5$ metabolic equivalents (METs), while in a sitting, reclining or lying posture [6–8]. The sedentary lifestyle of children has commonly been measured by the time spent in front of a screen [9–11]. A large-scale study conducted in 2015–2017 in 25 European countries on school-aged children (6–9 years old), reported that a mean of 25.2% of children spent 2 to 3 hours per day in front of a screen and 14.6% more than 3 hours per day [12]. The common sense census survey of the US reported that 8 to 12 years old children spent 4.44 hours per day on average using screen media; 25% spent 2–4 hours, 26% spent 4–8 hours and 15% more than 8 hours each day using a screen media, respectively [13]. French studies showed that more than 50% of school-aged children (6–10 years) spent more than 3 hours a day in front of a screen (ESTEBAN study) [14], and their screen time (ScT) has increased of 20 minutes per day between 2007 and 2015 (INCA2 and INCA3 studies, respectively) [15, 16]. However, sedentary time (ST) longer than 2 hours a day (evaluated mainly by time spent in front of TV) has been associated with an unfavorable body composition, reduced physical condition, lower self-esteem, lower sociability and poorer academic achievement in children and adolescents (5–17 years) [17]. In this context, designing and testing effective interventions to decrease SB in school-aged children therefore appears today to be a major public health issue (listed in the priorities of the French National Nutrition and Health Program) [18].

### The social gradient of sedentary

A social gradient has been identified in French children, with the educational level of the reference person in the household being inversely associated with the daily time spent in front of a screen (significant in girls and boys, and particularly marked in boys aged 6 to 10 years [8, 15]). Indeed, 65.5% reported spending 3 hours or more in front of a screen per day in

households where the reference adult had an education level below the baccalaureate (A-levels), compared to 34.7% among those declaring a baccalaureate (A-levels) or higher educational level (p<0.001) [8]. Moreover, recent studies suggest that a low parental educational level was unfavorably associated with screen use habits and could act as a moderator for certain factors associated with SB [19]. Thus, 6 to 10 years old children whose parents had shorter education were significantly more likely to use screen media more than 4 h/day [20]. Moreover, children aged 8 to 11 years whose parents had secondary education level or lower had significantly higher mean daily ScT than children whose parents had a university education level (i.e., 170.9 (±139.9) min/day and 125 (±112.2) min/day on average, respectively) [21]. Therefore, it seems important that the intervention to combat sedentary lifestyle takes into account the issue of social inequalities by considering the social context in which the individuals evolve.

## The relevance of a multilevel and participatory approach

Sedentary behaviors are determined by multiple influences. Working to reduce physical inactivity requires taking into account the individual in a holistic way in the context in which he/she evolves. Socio-ecological approaches defend the idea that effective prevention must take into account individual, social and environmental factors simultaneously and in a complimentary way [22–24]. As such, an intervention including modes of action at the individual level (behavioral mechanisms), but also at the social level (teachers, family), and at the environmental level (school, municipality, territory) appears particularly indicated. This type of multilevel intervention is all the better developed and adapted to the context if it is through a participatory approach, involving co-construction bringing together researchers, practitioners and representatives of the target population [25, 26].

## The trans-contextual model to explain individual's behavior

At the individual level, the effectiveness of a behavior change intervention is enhanced when an intervention is based on a theory [27]. The implementation of theory-based interventions has several advantages [28], including the identification of key psychosocial variables related to the targeted behavior, the selection of intervention techniques associated with these key psychosocial variables, as well as the understanding of the underlying causal processes explaining behavior change [29]. According to Flay, Snyder, and Petraitis [30], school-based interventions should focus on both personal (e.g., perceived control, self-determination), social (e.g., perceived norms, motivation to conform), and cultural variables (e.g., attitudes, values) to be effective in the school setting. From this perspective, a model such as the trans-contextual model (TCM) [31] has much to offer in the context of health promotion because it includes many of these personal, social and cultural variables. A meta-analysis by Hagger and Chatzisarantis to highlight evidence on the predictive aspect of self-determination theory on proximal predictors of intention and behavior, confirmed the relevance of the TCM for predicting health behaviors [32], and in particular regarding Physical Activity (PA) practice [33–35]. Attitudes, norms, and perceived control, the most proximal determinants of behavior in the TCM, were significantly associated with intention to remain sedentary [36, 37], suggesting the relevance of this model to predict SB.

## Previous interventional studies

A recent systematic review has investigated the effectiveness of multilevel interventions (i.e., involving consideration of at least two interventional levels) in reducing ST in children aged 5–12 years [38]. Thirty trials met the eligibility criteria and were included. Among studies

targeting 2 (n = 2), 3 (n = 19) and 4 levels (n = 9), 1 (50%), 9 (47%) and 7 (78%) were effective and reported significant reduction of ST, respectively. Therefore, interventions tend to be more effective when the strategies used are deployed among four levels. Thus, it seems that targeting four levels including intrapersonal, interpersonal, organizational and community, tend to lead to more successful interventions to reduce SB. However, among interventions aiming at reduce ST in children considered in two recent reviews [38, 39], only two using a multilevel framework based on the socio-ecological model (SEM) were implemented and evaluated in French children [40–42].

Moreover, if the theoretical anchoring on a model of behavior change has been put forward as an important element for the effectiveness of the intervention at the individual level [27], among the previous studies targeting the sedentary lifestyle of children, very few mentioned a theory, and when they did [43–45], the quality of the implementation of the theory was poor (i.e., the link between the intervention techniques and the theoretical variables was not explicit). While the TCM [31] has shown its relevance for predicting PA behavior [33–35], to our knowledge, it has not yet been tested to explain SB.

The CIPRES intervention aims to reduce ST in school-aged children (8–10 years old). Main original characteristics of the CIPRES intervention are: (i) to be a co-constructed intervention by using a participative approach involving key local actors (i.e., children, families, teachers, associative and municipality staff, and representatives of the French National Education), (ii) to use a multilevel implementation approach using four levels based on the SEM [46], (iii) to be based on the TCM to identify motivational levers of behavior change [31] and, (iv) to be assessed with a cluster-randomized controlled trial methodology and objectively measured ST by using accelerometers. This paper aims to describe the protocol of the CIPRES study ("Co-construct, Implement and evaluate a multi-level Intervention to PREvent Sedentary in children").

## Methodology

A written ethical approval was obtained from the ethics evaluation committee of Inserm, and the Institutional Review Board (IRB00003888,IORG0003254, FWA00005831) of the French Institute of medical research and Health.. Written informed consent was obtained from each participant and they were informed of their right to withdraw at any time without reasons.

### Study objectives

The primary objective of this study is to evaluate the effects of the CIPRES multilevel intervention on ST of children attending primary school (8–10 years old).

The secondary objectives are to measure the effects of the CIPRES intervention on children's ScT, PA, quality of life (QoL), body mass index (BMI) and socio-cognitive variables of the TCM (i.e., autonomous motivation, attitudes, norms, perceived behavioral control, intentions).

The primary hypothesis of the study is that the CIPRES multilevel intervention will decrease the ST of children. As secondary hypotheses, the CIPRES intervention should reduce children's ScT and BMI, and improve children's PA, QoL, and socio-cognitive variables of the TCM in favor of the reduction of SB.

### Study design

The intervention will be evaluated by a two-armed prospective cluster-randomized controlled study. Randomization will attribute either the intervention group (i.e., receiving the CIPRES intervention) or the control group (i.e., no intervention), considering municipalities as clusters

(i.e., all the classes of the same municipality belong to the same group) to avoid contamination between groups. The target population is made up of 700 French children enrolled in primary school (year 4 and 5) located in 4 municipalities in the Var department, southeast of France (i.e., Toulon, La Seyne-Sur-Mer, Hyères, Brignoles) (Fig 1).

The study includes two phases: 1) co-building of a SB prevention intervention using a participatory approach, and 2) implementation and evaluation of the intervention.

Phase 1: The SB prevention intervention was co-build by using a participatory approach including representatives of each "level" of the SEM (i.e., children, parents, teachers, pedagogical advisors, staff from associations and municipalities). Volunteers from each level of the SEM were proposed to join an advisory committee to create an intervention with a participatory approach. Researchers were asked to propose scientific levers (e.g., multilevel approach, school environment, motivation-related variables and potential behavior change levers based on the TCM) and the other participants (i.e., children, parents, teachers, pedagogical advisors, staff from associations and municipalities) brought field experience and propositions regarding forms of action and adaptation to the context [47].

Phase 2: School and classes recruitment and implementation of the intervention started the 1st of September 2021 and is currently on going. In line with the French education system's timetable, the intervention must last a minimum of six weeks, i.e., the time between two school vacation periods, in each recruited class and aims to reduce SB at different times in the child's life: school, extracurricular and home. An evaluation of the effectiveness of the intervention is carried out before (T0) and after the intervention (T1: T0 + 6 weeks). Children's ST is measured during everyday lifetime by accelerometers worn during seven days. Other variables such as the level of PA, declared BMI, ScT habits, and variables of TCM (i.e., autonomous motivation, attitudes, norms, perceived control, intentions) are also measured (Fig 1).

Since the socioeconomic characteristics of children and their parents have been associated with children's SB and PA in previous surveys [8], the deprivation level of each school location

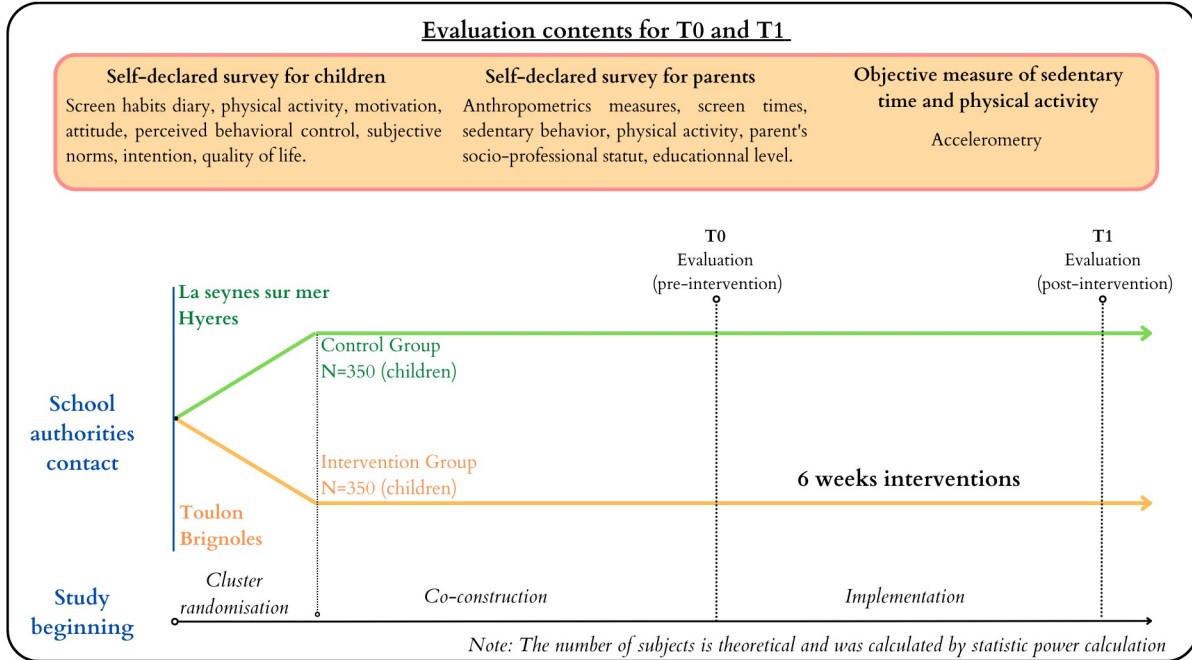

**Fig 1. Design of the CIPRES cluster randomized controlled trial.**

is collected. The Ecologic Deprivation Index (EDI) measures the level of deprivation at the smallest French geographical units [48]. The EDI is divided into quintiles (Q), from the lowest level of deprivation (Q1) to the highest (Q5). In France, children must attend the closest school to their home. So, the use of school EDI classification also provides information on the level of deprivation of pupils' families. Thus, EDI will be used to investigate the effect of the intervention according to the level of deprivation of the school location.

## Population

The targeted territories are four cities from southeast France next to the Mediterranean Sea (Var department) randomly assigned to intervention group (Toulon, Brignoles) or control group (La Seyne-sur-Mer, Hyères). The target population is made up of 700 children attending primary schools in these municipalities in the years 4 and 5 classes (aged 8 to 10 years old). This study will more broadly include the participation of key actors who are in contact with children or likely to play a role in the organization of school, extracurricular and family time for children, namely teachers (school time), educators including association staff and municipal staff (extracurricular time), and the parents of children (family time). The schools will be selected with the help of pedagogical advisors of the French National Education and then be contacted by the Health Education Committee of the Var department (the CODES 83). Schools and then classes will be recruited within the included municipalities, on a voluntary basis. For each class willing to participate in the study, parents receive a written note which includes explanations about the research's aims, the assessment methods (i.e., self-administered questionnaires for children and their parents, and accelerometers worn by children for 7 days at T0 and T1), and that participation is based on their own volition. Children are given consent forms to take home for their parents to sign. Parents have the choice of ticking three types of consents: (i) parents' consent to filling in the questionnaires; (ii) parents' consent to children filling in the questionnaires; (iii) parents' consent to children wearing accelerometers. Only children whose parents have given their consent will be included (Fig 2).

## Intervention

**Rationale and theoretical framework.** The CIPRES research project aims to develop an original interventional approach involving a double theoretical anchoring: (i) SEM taking into account influences of the social environment on individuals sedentary related habits and behavior [46], (ii) the TCM integrating motivational levers likely to facilitate behavior change (i.e., reduction of ST) [31]. The intervention will be based on three main principles: (1) Institutional participation is voluntary; (2) Based on the SEM [46], the intervention will involve a multilevel design including the participation of National Education professionals and teachers (school time), educators (extracurricular time), and children's parents(family time). (3) The intervention will be based on the TCM [31] in order to activate motivational levers for the reduction of SB.

*A multilevel approach with the socio-ecological model.* This last decade, there is an increasing amount of interventional research aiming at reducing children's ST [49–51]. These intervention strategies often focus on goal setting, motivation and intra-individual factors, mainly according to the social cognitive theories [50–53]. However, to be effective, public health interventions have to consider health behaviors in the light of the contexts in which they take place, and not neglect the broader environment. Indeed, sedentary lifestyle, like other health behaviors, are not only influenced by intention and rationality (e.g., knowledge, attitudes toward sedentary lifestyle). Therefore, creating an effective intervention targeting SB reduction requires action on intrinsic (e.g., motivation) but also on contextual factors, by jointly

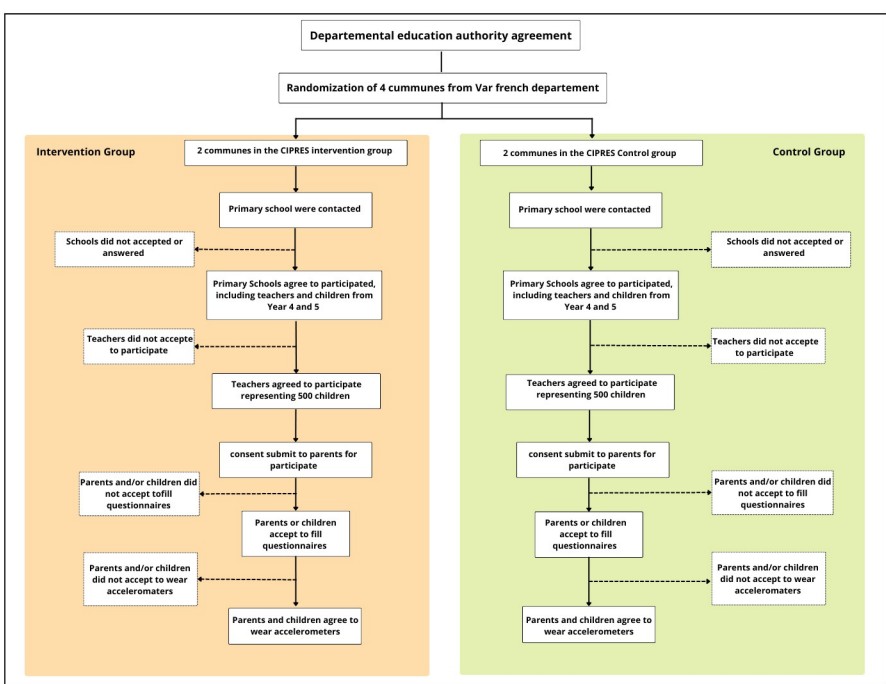

**Fig 2. Planned flowchart.**

implementing strategies at the meso and macro-environmental level [24, 54–56]. Thus, in this perspective, a theoretical anchoring on the TCM [31] allows to identify the motivational levers of SB, but it is relevant to integrate this model in the broader perspective of the socio-ecological approach.

In this purpose, the SEM, based on the original work of McLeroy and colleagues [46], provides a complex and powerful framework for public health interventions. It goes beyond the cognitive behavioral-based approaches, by considering intrinsic determinants and the social mechanisms of the production of health issues [57].

The ecological perspective allows identification of personal and environmental levers for health promotion and education interventions [58] with two main ideas: behavior is viewed as being affected by, and affecting, multiple levels of influence, as it can be graphically represented as a series of concentric circles in the Fig 3. Five levels of influence for health-related behaviors have been identified: (1) intrapersonal, or intrinsic/individual factors; (2) interpersonal factors; (3) institutional, or organizational factors; (4) community factors; and (5) public-policy factors (society scale). The second key point relates to reciprocal determinism between individuals and their environments, meaning that behavior both influences and is influenced by the social context [46, 58, 59].

Applied to the determinants of SB, this multifactorial approach states that these behaviors can be influenced by a multiplicity of levels, from the most proximal (e.g., self-esteem, intention, motivation, social support from family) to the broader settings (e.g., school environment, public policies). A summary of levels, examples of determinants and corresponding strategies of the CIPRES intervention is proposed in the Table 1.

Ultimately, in the socio-ecological perspective, a co-construction of the intervention or a participatory-based design with the targeted group is, more than a relevant strategy, a crucial component of the intervention paradigm [24], which has been done for the construction of the

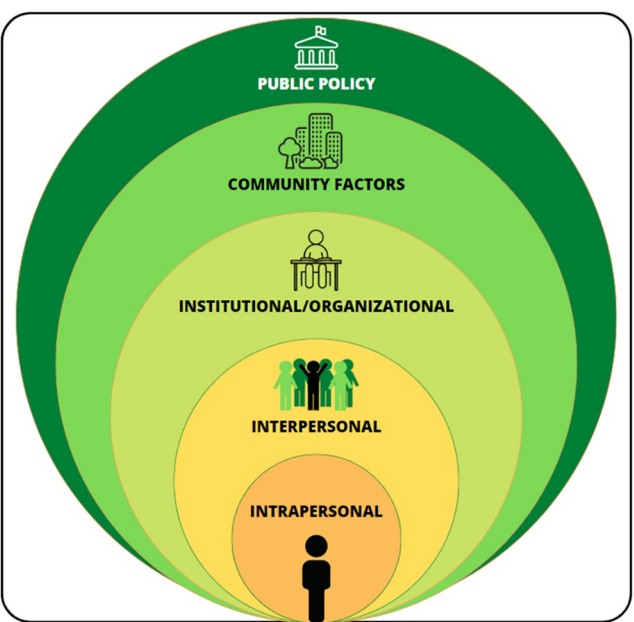

**Fig 3. Graphical representation of the generic socio-ecological model, adapted from McLeroy and colleagues (1988) [46].**

CIPRES intervention (see below, phase 1 in the 2-phase process). Moreover, in line with social health inequities concerns, as middle and upper-class individuals benefit the most from health programs (e.g., in France, the PNNS program) [18], the systemic ecological and the participatory perspectives may be more appropriate for disadvantaged populations and then to reduce social inequalities in health [60, 61].

*The trans-contextual model.* The effectiveness of a behavior change intervention is enhanced when intervention is grounded in a theory [27]. The TCM [31] is the integration of self-determination theory (SDT) [62]–a holistic theory of motivation—and the theory of planned behavior (TPB) [63]–a sociocognitive theory of behavior change. As shown in Fig 4

**Table 1. SEM applied to SB (inspired by Kellou et al., 2014) [54] and strategies of CIPRES intervention (examples by level).**

| Type of level | Examples of determinants | CIPRES interventional strategies |
|---|---|---|
| **Intrapersonal** | Psychological (e.g., knowledge about SB, intrinsic motivation) and physiological characteristics (e.g., health) | provide support for perceived autonomy to adopt healthy behavior. improve positive attitudes and subjective norms regarding SB; improve perceived behavioral control on the decrease of ST; increase intentions to adopt a less SB; informational and behavioral strategies (e.g., school thematic activities); Increase knowledge about SB-PA and benefits from ST reduction (e.g., public health recommendations). |
| **Interpersonal** | Social support of caregivers (parental rules, peers' behavior, encouragement from teachers. . .) | Target and involve family, peers, teachers and educators; increase norms favorable to the reduction of ST in school & family context (e.g., informational strategies for parents and family activities to reduce ST in family time booklet) |
| **Organizational** | Home; institution (care center, school), with physical and social dimensions, e.g., school wellness policy, garden equipment | Target various settings during school time (activities related to school subjects), extracurricular time (modules offering extracurricular activities), family time (modifying the organization of family leisure time by proposing modules). |
| **Community** | Neighborhood, community environment (e.g., local associations) | Punctual events at a local level (municipalities, local associations) |
| **Public policies / macro-environment** | Laws, national and local regulations (e.g., transport system, media, sports facilities in the city) | CIPRES intervention will not target that SEM level. |

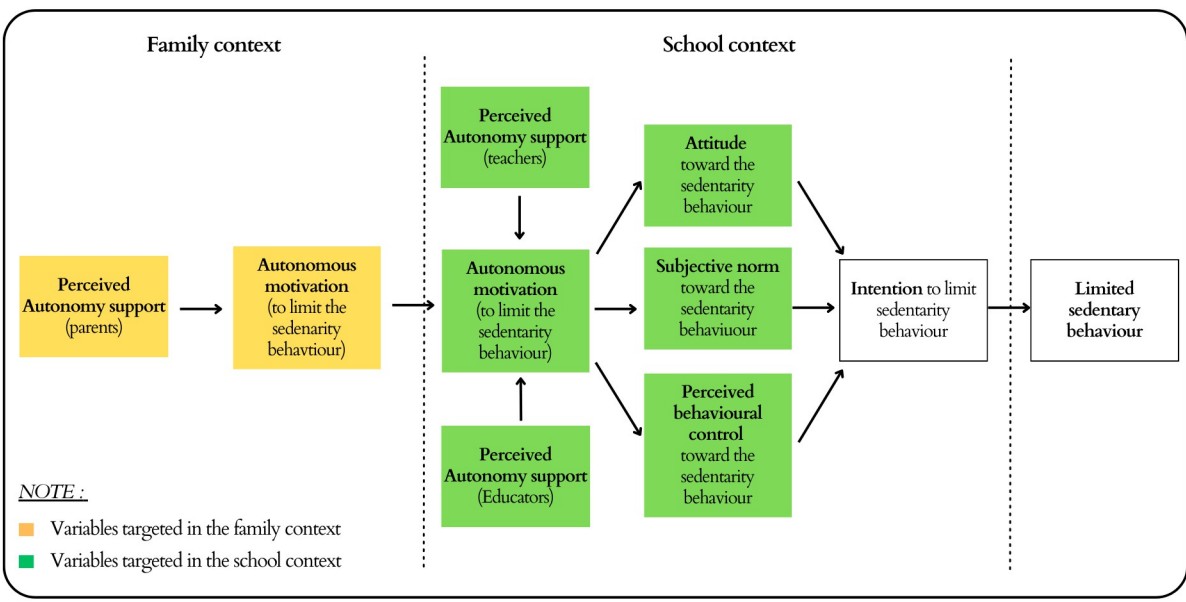

**Fig 4. TCM model in the CIPRES study (adapted from Hagger et al., 2009) [34].**

this integration implies that the SDT variables (e.g., perceived autonomy support, and personal motivation) explain part of the sociocognitive concepts of the TPB (e.g., attitudes, norms, behavioral control). More specifically, the TCM proposes that perceived autonomy support (i.e., giving choice, providing meaningful arguments in favor of adopting a behavior, limiting pressure to adopt the behavior, and encouraging personal initiatives) and autonomous motivation (i.e., performing tasks and behaviors out of a sense of choice, ownership, and personal agency) of SDT are related to social and cognitive constructs that influence TPB variables such as attitudes (i.e., favorable or unfavorable evaluation of the target behavior), subjective norms (i.e., perceived social pressure from significant others on performance of the behavior, and level of adoption of the behavior by significant others), and perceived behavioral control (i.e., perceived ease of performing the behavior).

In their book Hagger and Chatzisarantis [31], suggest that TCM is made of three theories of motivation to explain the motivational process influencing motivational transfer to the intention to take action and could explain change in SB. On one hand, the TCM is based on the postulate that SDT and TPB can offer complementary explanations for the adoption of health behaviors i.e., belief-based judgments in the TPB would depend on the motivational orientation described by the SDT. On the other hand, the Vallerand's hierarchical model [64] provides an explanation about the link between the SDT's autonomous motivation and the TPB's intention to adopt behavior. It has been proven in particular with regard to the promotion of PA [33–35] but, to our knowledge, it has never been tested on the reduction of SB. The CIPRES program considers two complementary ways of action: (i) as suggested by the SDT [62], an intervention based on the TCM should focus on the motivational climate, in particular by supporting children's autonomy, a privileged role both for parents and for teachers and educators; (ii) as suggested by the TPB [65], an intervention based on the TCM should also focus on conveying messages targeting salient beliefs related to children's attitudes, subjective norms and perceived behavioral control to influence the intention of adopting a new behavior, a particularly important role for teachers and educators [66]. The implementation of the

CIPRES intervention will include the participation of both educational personnel (e.g., teachers, extracurricular educators) and parents.

**Two phases process: Co-construction and implementation.** A two-phase process has been planned regarding the creation and the implementation of the CIPRES intervention:

*Phase 1*: *Co-construction of the intervention based on a participatory approach.* The preliminary following guidelines have been identified to guide the co-construction of the CIPRES intervention according to SEM, TCM and previous published interventional studies [38, 39]:

1. Identify "gate keepers", key individuals who open access to the field.

2. Involve multilevel relevant actors for the co-construction among children, parents, teachers, pedagogical advisors and educators.

3. Involve multilevel relevant actors for the implementation of the intervention (i.e., teachers, parents, and educators)

4. Increase autonomous forms of motivation to reduce SB in school and family contexts.

5. Increase attitudes, norms and perceived behavioral control favorable to the reduction of sedentary in the school context.

6. Propose PA or sedentary breaks to reduce SB during school hours, extracurricular time and family time.

A participatory process has been used for the co-construction of the intervention, involving children, parents, teachers, pedagogical advisors and educators. All the actors were invited to take part in the co-construction at the same time frame, the meetings were spread over several sessions. These multilevel actors were asked to share their experiential knowledge about children sedentary and the levers/brakes regarding the implementation of prevention activities/actions, considering the field context in which they evolve. This participatory approach has been chosen in order to build an intervention in the most adapted form according to the context and the possibilities of implementation in the included territories. Staff from the French Ministry of Education who were present when the project was set up and CODES-83, both identified relevant field actors from the two interventional communes. Second, these actors were contacted by email, telephone or direct contact to propose they participate in the project; a brochure presenting the CIPRES project and the co-construction process was distributed. Third, an advisory committee (working group) was constituted with field actors who wanted to participate, in addition to researchers. Researchers and field actors, including parents, teachers and educational staff, pedagogical advisors, elected municipal officials and educators, met regularly for 6 months. These gatherings were implemented via remotely (zoom) or physical modalities whenever possible. In summary, the modalities of discussions were in person interviews (n = 6) and physical meetings (n = 6); remote/distance meetings (n = 6) and interviews (n = 3); and participatory observation and intervention (n = 3). By involving a broad range of actors (children, parents, municipal leaders, departmental and local educational advisers, teachers, school directors, school nurses, and after school staff, associative coordinators, Ph.D. student and national education trainee), several levels of intervention, separately or combined, were targeted by the community (e.g., associations, leisure time), organizational (e.g., school, home, after-school) levels, and also the micro-environment of the child (parents and caregivers).

These exchanges allowed a better understanding of the context and the targets of the intervention, in particular for territories marked by severe socio-economic deprivation. Moreover, this co-construction stage allowed us to identify brakes, levers and key strategies mentioned by

stakeholders (e.g., Putting physical activity back into the school curriculum or more broadly, into education. Information session for parents in the presence of the children. Involve parents via their children's activities. Use local players to mobilize parents. Include intervention modules in the school curriculum).

*Phase 2*: *Implementation in the context of the co-constructed intervention*. The intervention will last for 6 weeks in each class (duration between two periods of school holidays) and will aim to reduce SB at different times in the child's life: school time, extracurricular time and leisure time (e.g., time spent with family). Multilevel actors will be involved in the implementation of the intervention i.e., teachers, parents and educators. Compared to a pre-formatted intervention, the participatory co-construction of the CIPRES intervention prior to its implementation should facilitate its implementation through better involvement of the actors and better adaptation of the intervention to the context.

## Intervention content

The intervention contents have been created during the first phase (co-construction) based on the SEM [46] and the TCM [31]. With the participation of multilevel local actors, different media have been created in book form to reach multilevel actors of the SEM (educators, teachers, and parents) to provide a multilevel influence on children's SB. All the following intervention contents are made to be used during the experimental phase of the study. The different content will be provided to the intervention group for six weeks. The different booklets will be provided to all schools of the control group as well, but only at the end of the study.

**School time booklet.** The school's booklet presents six activities to be implemented by teachers in schools' classes and aimed to raise awareness among children about SB and decrease their ST as well as increase the perceived autonomy support. It is made up of fifty-six pages including study presentation, TCM presentation for teachers, guidelines for teachers to support children's autonomy, and the six activities that teachers are requested to implement in their class (i.e., Activity 1: "Sport, PA and SB definition table"; Activity 2: "Sportsman and sportswoman interviews"; Activity 3: "Space-time"; Activity 4: "Relay me if you can"; Activity 5: "Challenge day—no screen today!"; Activity 6: "Cooperative PA challenge"). The guidelines for supporting autonomy are based on the Hagger and Chatzisarrantis developments [34], and aim to provide autonomy-supportive teaching styles examples to improve the perceived autonomy support from teachers to children such as: do not use controlling language including words like "should" and "must", but prefer providing choice about types of non-sedentary activity; provide a personal initiative environment, enabling children to make choices, initiate activity or assume responsibilities (e.g., "What would be the best practice to be less inactive?", "what would you love to do after school instead of watching TV", "would you like to team build an activity for break time?"). Based on the co-construction of the intervention with teachers and pedagogical advisors of French National Education, all of the teacher's booklet activities have been associated with a school subject (e.g., math, civics, physical education, French) providing exercises and skills in accordance with the school academic program and at the same time, aiming to influence different motivational levers from the TCM. To facilitate the implementation, this booklet has been designed in ready to use form to be easily accessible and workable for teachers, with the participation of pedagogical advisors and project managers in health education. Every activity record comprises essential information including activity presentation, the associated school subject, pedagogical instructions, material needed, session duration, and targeted variables of the TCM (and annexes if needed). All the six activities have to be done once during the implementation time. All activities from the school time booklet are detailed in Table 2. A supplementary activity booklet has also been created to support

**Table 2. School time booklet activities.**

| Title | Skills worked | Group | Duration | Presentation | Material | TCM variables targeted |
|-------|---------------|-------|----------|--------------|----------|------------------------|
| Sport, PA and SB | Understanding how language works: acquire spelling lexical and enrich the lexicon | Entire classroom | 30 min | Pupils should learn about sedentary lifestyle, physical activity and sport by filling in a table. The teacher can present the attached "Definitions" sheet to the class beforehand. "Definitions" in the appendix. | Grey pencils Sheet I -II "Definitions" & "Activity Classification Table" | Attitudes |
| Sportsman and sportswoman Session 1 | Read—understand a literary text and make it their own | Entire classroom | 1 hour | Understand and analyse a sportsperson's interview. Each of the 5 "sportsman interview" sheets in the appendix contains a presentation, interview, lexicon and information. | Pens Sheets "Interviewing the athlete" | Attitudes Subjective norms |
| Sportsman and sportswoman Session 2 | Writing—using writing to reflect and to learn | Entire classroom (divide into groups of 4) | 1 hour | In this session the aim is to divide the class into groups of 4 and have them work independently on an interview sheet. | Pens Sheets "Interviewing the athlete" | Attitudes Subjective norms |
| Sportsman and sportswoman Session 3 | Read—understand texts, documents and images and interpret them | Entire classroom (divide into groups of 4) | 1 hour | Based on the work done in session 2, the students make an oral presentation in groups. | Pens Sheets "Interviewing the athlete" | Attitudes Subjective norms Support for perceived autonomy |
| "Space-time" Session 1 | Solving problems involving time. Solve problems involving simple conversions from one unit to another: relationships between usual units | Entire classroom | 1 hour | In this activity, students are asked to complete a schedule that allows them to spread out 7 hours of physical activity over the week. The recommendations for a child of this age are 1 hour of physical activity per day or 7 hours per week. This activity takes place over three sessions. The teacher gives students a pre-filled weekly plan of physical activities (active travel, PE, sports, etc.). Then the students have to solve the following problem: what is the total amount of physical activity performed during the week? To do this, they need to give the durations for each activity, which may involve converting units of time. Do they manage to reach the health recommendations for physical activity for a child? if not, how much activity time will it need to achieve to reach them? | Sheets Pens "Pre-filled weekly planner" | Subjective norms |
| « Space-time » Session 2 | Solving problems involving time. Solve problems involving simple conversions from one unit to another: relationships between usual units | Entire classroom | 1 hour | Students are asked to divide their usual physical activity time on the weekly schedule. Students are asked to calculate the total time spent and the deviation from the recommended 7 hours per week. Students have to complete the weekly planner with achievable activities to achieve these recommendations. It's important to suggest that students insert physical activities in place of screen time. The following week, the students must try to put into practice planned physical activities. a report will be made in session 3 on what has been achieved. | Sheets Pens "Weekly planner to complete". | Support for perceived autonomy Intention Perceived behavioral control |
| "Space-time" Session 3 | Solving problems involving time. Solve problems involving simple conversions from one unit to another: relationships between usual units | Entire classroom | 1 hour | During this activity, pupils draw up a report on the activities carried out during the last week. | Sheets Pens "Weekly planner to complete". | Support for perceived autonomy Perceived behavioral control |

*(Continued)*

**Table 2.** (Continued)

| Title | Skills worked | Group | Duration | Presentation | Material | TCM variables targeted |
|---|---|---|---|---|---|---|
| "Relay me if you can." | Developing motor skills and build a body language. mobilise different resources to act efficiently. | Entire classroom | 45 min | "The activity takes place in the playground. The teacher sets up a materialized course with the elements of his choice. The pupils run a relay race during which they collect images that they have to sort out (physical or sedentary activity). This activity takes place in two rounds. | It depends on the course set up (poles, hoops, etc.). "AP Images / Sedentary lifestyle". | Attitudes Perceived behavioral control |
| "Challenge day-no screen today." "Session 1" | Respecting others: respecting commitments made to oneself and to others | Entire classroom | 1 hour | This activity takes place in two stages and aims to replace as much screen time as possible with alternative activities (e.g., physical activities). The teacher challenges the students to spend a Wednesday with as little screen time as possible. Take time beforehand to discuss with the class when they use screens, with whom and what they use it for. | Pen "Roadmap." | Perceived behavioral control Subjective norms |
| "Challenge day-no screen today." "Session 2" | Respecting others: respecting commitments made to oneself and to others | Entire classroom | 1 hour | The children are asked to write on a sheet of paper what they did as a activity and how it made them feel. The next day after the challenge, discuss the effects felt, potential share with their families, as well as what each of them is doing to switch this screen time. | | |
| "Cooperative PA challenge" | Accepting differences: being able to cooperate | Entire classroom | 30 min | Pupils have to complete a cooperative challenge to run a half-marathon distance (21 km) together in 8 minutes. The distances covered by each pupil are added together. Each pupil runs at his/her own pace and can stop the activity at any time. There is no individual ranking as the aim is to encourage collaboration. | Plots Stopwatch "Tracking sheet" & "Half marathon diploma" | Subjective norms Support for perceived autonomy |

teachers who wish to implement supplementary activities. It is designed in the same way as the school booklet.

**Extracurricular time booklet.** The extracurricular booklet presents activities to be implemented by educators who take care of the children during meridian and after-school time. Its function is to strengthen the impact of school time booklets on the variable of the TCM and to raise children's awareness on the theme of sedentary lifestyles as well as increase the perceived autonomy support during extracurricular time. It is made of forty-pages including study presentation, guidelines for educators to support children's autonomy, and seven activities (e.g., "Debate game", "Swap me if you can", "10000 Step", "Picture me", "The game of the goose", "Fitness trail", "Giant memory", "Poster design"). The guidelines for supporting autonomy are close to the teacher's recommendations and aim to provide autonomy-supportive guidance styles examples to improve the perceived autonomy support from educators to children. To facilitate the implementation, this booklet has been designed with the participation of CODES 83 project managers in health education and will be provided to educators in ready to use form to be easily accessible and workable. All activities are made of essential information including activity presentation, pedagogical instructions, material needed, session duration. During the six weeks' interventions, educators have to implement as much as possible activities. All activities are presented in Table 3.

**Family time booklet.** The family time booklet was specifically designed for use in the family context, providing relevant information for parents and children on the topic of SB, PA, screen use and health-related consequences to complement and support the action of booklets

**Table 3. Extracurricular time booklet's activities.**

| Activity | Group | Duration | Objectives | Material | TCM variables targeted |
|---|---|---|---|---|---|
| Debate game | 3 teams | 30 min | The children are invited to discuss the notions of a sedentary lifestyle, ScT and PA. These notions must be preliminarily defined by the educators. | Sheets and pens | Attitudes |
| Swap me if you can | Individual or group work | 2x 30 min | Children are invited to think about alternative ways to reduce ScT, by swapping one screen activity with a non-sedentary activity of their choice. This activity is composed of two sessions. | Pens | Perceived behavioral control Autonomy support |
| 10000 Step | Whole group | 30 min | The aim is to make children realize what 10,000 steps represents in terms of distance / PA volume by doing the 10,000 steps co-operatively by groups. 10,000 steps represent approximately 6 km and correspond to the daily national recommendations. | Scrollable meter and studs | Attitudes Perceived behavioral control |
| Picture me | Individual work | 2x 1h | In this activity, children are asked to think about the amount of time they spend in front of screens through artistic activity and present their work to their pears and discuss/compare their ScT. | Grey pencils and Colored pencils, markers, paint, glue, scissors | Subjective norms |
| The game of the goose | Individual or in groups work of 2 to 3 | 1h | The goal of this game is to arrive first on the last square of the board. To achieve this, you must succeed in answering the various questions and challenges about PA. | counters and dice | Attitudes |
| Fitness trail | Groups work of 2 | 1h | In this activity, the educator asks questions related to SB / PA to the children who have to complete a sports course in order to pick up the right answer in a box. An example of a course is available in the appendix, but you are free to set up a sports course of your choice with the material you have. The course must be duplicated as many times as there are teams. | Material to create several identical courses | Attitudes |
| Poster design | 4 groups | 1h | This activity consists of creating a poster by theme: TV, sedentary, PA, guidelines. The goal is to prevent SB and promote PA, show the benefits and disseminate knowledge and information on this subject to other children. | Pencils, markers, paint, sheets | Attitudes Subjective norms |

used in the school and extracurricular context. It aims to promote interactions between parents and children in order to promote SB- and PA-related knowledge, family relatedness and choice and then favorably influence the targeted variables of the TCM in the family context (i.e., perceived autonomy support, autonomous motivation). It is composed of 24 pages including five different topics (i.e., scientific information, recommendations—advice—good habits, family activities, tools, web resources).

*"Scientific information"*. This chapter provides information on SB and consequences of SB in general. Families will find information on the health-related risks linked to a sedentary lifestyle (e.g., chronic diseases), the thresholds of physical inactivity, scientific data about children ST in France, and the health-related benefits of PA.

*"Recommendations, advice and good habits"*. This section provides recommendations about the healthy levels of PA for both children and adult [67], as well as suggestions/ideas for decreasing and/or break ST and screen use with their children in everyday life (e.g., break sedentary lifestyle regularly, find alternatives to screens to keep busy, creates rules for ScT). It also provides healthy lifestyles advice to prevent negative impact of ScT (e.g., four-time days with no screen to develop child autonomy, language, thinking, imagination, ability to be alone) [68] and tips on how to increase weekly PA, such as taking the stairs, riding a bike or getting up from your desk regularly.

*"Family activities"*. The booklet intends to favor family relatedness by proposing three SB prevention activities to be done together as a family, each lasting an average of 15 minutes: *i*) "My family and screens", is an awareness-raising activity which aims to enable the various

members of the family to realize how much time they spend in front of screens; *ii*) "Active break", aiming to propose an active challenge during ST (e.g., during TV advertisement) to be done as a family every evening or weekend. Propositions of challenges are provided in the booklet appendix (e.g., dancing, doing squats, yoga, etc.), families can choose one. *iii*) "A task for all", the aim is to assign a household task to prevent SB during family time to each member of the family for one week. Each week a rotation is proposed so that each member can participate in all tasks.

*"Tools"*. The tools chapter lists several numeric applications that can be used to limit ST or ScT while encouraging PA practice during family time.

*"Resources"*. The resource section includes links to the websites used to create the booklet as well as useful links for parents who can access relevant sources of information regarding SB, ScT, physical inactivity and PA.

## Evaluation

**Effectiveness assessment.**   Intervention effectiveness will be assessed by comparing the intervention group to the control group at two-time points, T0: pre-intervention (baseline) vs T1: post-intervention (T0 + 6 weeks, the end of the intervention i.e., the time between two school vacation periods). The primary outcome will be the ST of children for 7 days (i.e., including weekdays and weekends days) assessed by accelerometer.

In addition, several secondary outcomes will be measured at two-time points:

(i) The children's time spent in front of a screen reported by children's parents following three categories: a) Watching time including TV, web videos or time spent in front of a TV during lunch time, b) Gaming time including gaming time on console, smartphone, tablet and computer. c) Other ScT including homework, social media or texting in front of a computer, smartphone or tablet.

(ii) The PA level of children assessed by accelerometer and parents' questionnaire.

(iii) The motivational and sociocognitive variables of the TCM (i.e., autonomous motivation, attitudes, norms, perceived behavioral control, intentions) measured by questionnaires administered to children

(iv) BMI of children (according to weight and height of children declared by parents).

(v) Quality of life assessed by self-reported questionnaire administered to children

**Children PA and SB objective assessment.**   Commonly used for objectively measuring PA, Triaxial GT3X+ accelerometers (ActiGraph,FL,USA) will be used to assess ST and different levels of PA (i.e. low physical activity (LPA), moderate physical activity (MPA) vigorous physical activity (VPA)) and moderate to vigorous physical activity (MVPA)) for children. The devices will be placed on the dominant wrist during 7 consecutive days, and 24 h per day. Nevertheless, children will have to remove it when they go to a water environment (e.g. bath time, shower, swimming. . .). Accelerometers will be initialized to record data at a sample frequency of 60 Hz using ActiLife software (V6.13.4 Full Edition, ActiGraph, FL, USA). Non-wear time will be defined using Choi's et al., algorithm which consists of 60 min of 0 counts per minutes (cpm) with an allowance of 2 min of activity when it is placed between two 30 min windows of 0 cpm [69]. Validated wear time's data will be downloaded and aggregated into 5s epochs and daily levels of ST and PA will be determined using Crouter's et al., validated cut points [70].

**Children self-reported screen habits and school activities.** A diary will assess children's habits of screen use. Children will have to report their favorite screen (e.g., smartphone, TV, computer, video game or tablet), which one they use the most, if they watch screens in their bedroom, where they spend the most of the time in front of a screen (e.g. bedroom, kitchen, living room, outdoor, in a friend's house) with who (e.g. family members or friends), and what kind of content (e.g. social media, movies/series, cartoons, video games or documentary). The diary will also allow children to report what PA type they do at school with their teacher, how long and how often as well as what kinds of activities they do during break time (e.g., board games, ball games, hopscotch, talking, catch me if you can or others).

**Psychosocial variables in children.** The TCM variables will be measured with two self-reported questionnaires administered to children in class including questions related to self-determined motivation, attitudes, norms (subjective and descriptive), perceived behavioral control and intentions. Each survey will be evaluated on its own scale.

The study will use a 15-item survey which was created to assess the motivational level for spending less time in front of a screen. Children will answer the questions, all starting with "I spend less time in front of screens because. . .", using a five-point scale: 1 [not true]; 2 [a bit true]; 3 [quite true]; 4 [True]; 5 [very true]. The scale includes five motivational dimensions, each one being measured with three items: intrinsic motivation (e.g., "because I don't like watching TV"), identified regulation (e.g., "because it's important for my health"), introjected regulation (e.g.,"I feel guilty if I spent too much time in front of a screen"), external regulation (e.g. "to please my parents"), and amotivation (e.g.,"I don't understand why we need to spend less time in front of the screens"). A Cronbach Alpha coefficient will be used to test the correlation between items from the same dimension.

The assessment of other psychosocial variables of the TCM (i.e., attitudes, norms (subjective and descriptive), perceived behavioral control and intentions) is based on current guidelines [71] and will be based on a 16-item survey using a four-point scale to respond: 1 [not at all]; 2 [not really]; 3 [yes, a bit]; 4 [yes, a lot]. Attitudes will be measured using 3 items (e.g., do you think that spending a lot of time in front of a screen will be boring?). Perceived behavioral control for children to decrease screen use in leisure time will be assessed using 3 items (e.g., do you feel able to easily find some activity to do without using screens?). To assess descriptive norms, the questionnaire will include 3 items (e.g., does your mother spend a lot of time in front of screens in her everyday life?). Subjective norms assessment will be based on 4 items (e.g., does your teacher want you to spend less time in front of screens?). Finally, intentions will be measured using 3 items (e.g., during the next week, will you want to spend less time in front of screens?).

**Children quality of life.** The French translated form of the kidscreen-10 questionnaire will be used to assess the quality of life in children. Kidscreen-10 is a validated self-reported health-related quality of life questionnaire for children and adolescents aged from 8 to 18 years, derived from the Kidscreen-27 itself derived from the Kidscreen-52 [72, 73]. It is made of ten questions using a five-point scale to respond: 1 [never]; 2 [sometimes]; 3 [often]; 4 [very often]; 5 [always] excepted for questions 1 and 9, which use another five-point scale to respond: 1 [not at all]; 2 [a little]; 3 [moderately]; 4 [a lot]; 5 [extremely].

**Evaluation of children behavior by their parents.** This study will use an adapted version of the SIT-Q-7d questionnaire that has been validated in adults [74]. This parent's survey is a self-reported survey that quantifies their children's behavior in the last 7 days regarding SB and PA across several domains covering the children's daily life activities:

(i) Children's sleep habits: Parents will be asked to report the average wake up time and bed-time of their children for the last seven days during weekdays and weekend days, out of the

holidays. It will exclude the time spent in front of a screen before sleeping or after waking up. In addition, it will be asked to report the average time spent napping for the last seven days during weekdays and weekends following a six-point scale: 1 [No nap]; 2 [1–15 min]; 3 [15–30 min]; 4 [30–45 min]; 5 [45 min-1h]; 6 [more than 1h].

(ii) Children's SB: Parents will be asked to report their child's sedentary habits, and more precisely, time spent in front of a screen, sitting activities or lying down activities, on the last seven days out of holidays. About ScT, parents will be asked to declare the time spent by their children: (i) watching TV (including TV, YouTube or other media platform, DVD, blue ray) during or out of mealtime, (ii) playing video games (including computers, games console, tablet or smartphone) without including standing games, and (iii) using a computer, tablet or smartphone for other activities (including homework, social media, SMS or internet surfing). All questions are scored following a nine-point scale: 1 [never]; 2 [1-15min]; 3 [15–30 min]; 4 [30 min-1h]; 5 [1h-2h]; 6 [2h-3h];7 [3-4h]; 8 [4h-5h]; 9 [more than 5h]. About other sedentary activities, parents will be asked to declare time spent sitting or lying down by their children without using a screen, such as doing their homework, reading a book or taking leisure time (e.g., board games, drawing, listening to music, etc.). All questions are scored following an eight-point scale: 1 [never]; 2 [1-15min]; 3 [15–30 min]; 4 [30 min-1h]; 5 [1h-2h]; 6 [2h-3h]; 7 [3h-4h]; 8 [more than 4h].

(iii) Children's PA habits: Parents will be asked to report the time spent by children in PA on a typical week, out of the holidays. This section is made of two tables, one refers to the leisure PA done in a sports club, and the other concerns the leisure PA done during the family time out of a sports club. Parents can declare several activities in the two categories; for each declared activity, they are asked to report: (i) the average time spent in the activity using a five-point scale (1 [less than 30 min]; 2 [30 min-1h]; 3 [1h-1h30]; 4 [1h30-2h]; 5 [more than 2h]); (ii) the week frequency using a five-point scale (1 [less than 1 time a week]; 2 [1 time a week]; 3 [twice a week]; 4 [3 times a week]; 5 [more than 3 times a week]); and (iii) intensity using a three-point scale (1 [light]; 2 [moderate]; 3 [high]).

(iv) Children's travel habits: Parents will be asked to report sedentary and active transport habits of their children. This section is made of two tables. The first table concerns time spent between house and school and the second table includes the time spent to go to other activities/locations. Each table is divided into two ways of transport, motorized (e.g., car, bus, train, etc.) and non-motorized (e.g., walk, bike, skateboard, etc.). The average transport time during the last seven days (out of holidays) is requested to use a nine-point scale: 1 [never]; 2 [1–15 min]; 3 [15–30 min]; 4 [30 min-45 min]; 5 [45 min-1h]; 6 [1h-1h30];7 [1h30-2h]; 8 [2h-2h30]; 9 [2h30-3h].

(v) Family social characteristics: Characteristics of respondents to survey (father, mother, other. . .) are asked, including the birthday of each parent, and some information about their social situation: homeowner (yes/no), number of children in care, if children have TV in their room (yes/no), the socio-professional category and the study level of both parents.

**Implementation process assessment.** As suggested by Hawe *et al*., we will assess how the intervention was implemented in the context [75]. Follow-up sheets for the "sessions" carried out will make it possible to count the number of sessions and the participation of all the actors. Semi-structured interviews will be carried out at the end of the intervention with representatives of actors that have participated in the implementation of the intervention (e.g., teachers, associative staff, project managers). These interviews will make it possible to obtain qualitative

information on how the study was implemented, and what the facilitators/breaks were in the implementation. In addition, an intervention monitoring grid will make it possible to identify the achievement (done/not done, why?) of the intervention modules; an intervention monitoring sheet will be drawn up including all the sessions scheduled for the teachers and associative staff and will be completed and monitored by the project manager who will centralize the information.

**Feasibility study.** One preliminary pilot study has been conducted to confirm that schools and classes recruitment was feasible and that to check the participating children and parents understood and answered the evaluations. This pilot study targeted 111 children from two communes of the Var department (Toulon and Laroquebrussane). Children were aged from 7 to 11 years old attending four public schools. Among the targeted population, 100 children and their parents accepted to participate resulting in 90% of the acceptance rate. We found that the procedure for distributing and collecting accelerometers was feasible, with 78 children and their parents giving their consent for the children to wear the accelerometer for seven consecutive days (i.e., 78% of acceptance rate). We also confirmed the feasibility of the parents' questionnaire, with 64% of the completion rate. In class, the administration of children's questionnaires was feasible (evaluating lifestyle habits, TCM variables and quality of life) resulting in approximately one hour for the completion of the questionnaire for a class of 20–25 children.

## Statistical considerations

**Sample size calculation.** In a French study measuring ST with Actigraph GT3X accelerometers, children aged 7 to 10 accumulated 583 minutes on average (standard deviation: 57 min) of ST per day [76]. We will aim with the CIPRES intervention to reduce by 30 minutes per day the ST in the target population. To evaluate the number of subjects needed, we will choose α risk equal to 5% and β risk equal to 10% (power = 90%). Without considering clusters, showing a decrease of 30 minutes of ST per day between pre- and post-intervention will require to include 92 children per group (taking into account a correlation of 0.4 between baseline and follow-up measures). Randomization by cluster requires taking into account inflation factor of our sample size because of the homogeneity of the conditions within each cluster: $1+(m(1+cv^2)-1)r$ (m being the average number of subjects per cluster, cv a coefficient of variation equal to the ratio between the standard deviation of the size of the clusters per m, and r being the intraclass correlation coefficient). In our study, the cluster is a municipality. With a potential average of 150 children included per municipality, a coefficient of variation at 0.5, and a rho coefficient set at 0.01 according to databases of intraclass correlation coefficients [77] and a recent French study [78], the inflation factor is equal to 2.87. The number of subjects needed to be able to show a difference of 30 minutes per day in ST is estimated at 527 children. Taking into account a potential rate of 25% of lost-to-seen over time, the number of subjects to include in the CIPRES study amounts to 700 children (i.e., 350 in the intervention group and 350 in the control group), corresponding to approximately 14 classes per group (i.e., approximately 7 schools per group).

**Statistical analysis.** All the data will be electronically processed. Original study questionnaires and forms will be entered and kept at the coordinator site (University of Toulon) in a secure and accessible place. After quality control of the database, statistical analysis will be performed.

Baseline comparisons between arms will be performed using univariate analysis of variance for continuous variables and chi2 test for categorical variables. An intent-to-treat analysis will be done. ST of the CIPRES interventional group will be compared to ST of the control group

by using mixed-model analysis which allows assessing changes over time between groups within a repeated measures design with the determination of time effect (T0, T1), group effect (CIPRES, control) and, the interaction effect between time and group. Group, time and group-by-time interaction will be treated as fixed effects, and schools, classes and individuals will be considered as random effects. The analysis will be repeated on the secondary endpoints, using a linear mixed model for the continuous variables. Unadjusted and adjusted analyses controlling for potential confounders (e.g., age, sex) will be performed. In addition, a per protocol analysis will be carried out according to the adherence of classes to the intervention.

**Implementation process analysis.** Quantitative indicators (e.g., acceptance, attendance, attrition rate, proportions) will be described and compared between the schools included and will provide information on acceptability and implementation. The textual data available from the interviews with representatives of actors that have participated in the implementation of the intervention (e.g., teachers, associative staff, project managers) will be processed by thematic analysis. The analysis will reveal the frequencies and the strengths of the text (e.g., compliance / non-compliance, levers / brakes, organization, composition of the teams, school dynamics, etc.) that could have an impact on the feasibility of the proposed intervention model. A mixed method approach searching convergence between quantitative and qualitative results will be favored for the analysis [79].

## Discussion

Therefore, in view of the state of the art, the proposed project to reduce the ST of school-age children (8–10 years) presents several originality and scientific innovations:

1. Design and implement a co-constructed intervention with a participatory approach, involving several levels of key local actors and allowing an adaptation to the context,

2. This intervention is based on a SEM (i.e., including modes of action at the individual, social and environmental levels) and will include a multilevel approach,

3. This intervention is grounded in a theory of behavior change (i.e., the TCM).

4. The question of social inequalities is raised in this project, in particular by the characterization of the level of deprivation of the schools locations (by using the EDI) [48] which will allow the comparison of several distinct areas of intervention according to the level of deprivation.

5. This intervention will be the subject of a process analysis based on the FIC model (key function, implementation, context) [47], with a view to its transferability, so that deployment in different contexts can be envisaged.

In the meta-analysis of Rodrigo-Sanjoaquín et al., among the school-based interventions of acceptable methodological quality recently identified to reduce ST measured by accelerometer in school-age children [39], nine interventions targeted several levels of actors (i.e., child, teachers, parents, educators), only three were based on the SEM [40, 80, 81], only one has been designed and implemented in France [40] and none study was anchored on the TCM. The CIPRES study will provide scientific elements regarding the obstacles and levers to the implementation in France of a theoretically anchored intervention aimed at reducing the sedentary lifestyle of school-age children, involving multiple levels of actors (i.e., child, parents, educational staff, municipal staff) using a socio-ecological approach [22–24], and based on a behavior change model i.e., the TCM [31]. Thanks to the theoretical anchoring of the intervention on the TCM, this study will shed light on the motivational mechanisms at play. The concepts

of the TCM studied will inform us about the role of beliefs in the reduction of risky behavior such as SB. According to this model, the synergy of the role of educational staff and parents in supporting autonomy will create a climate of motivation favorable to reflection on sedentary living in the family and at school (i.e., support for autonomy and development autonomous forms of motivation) and will promote the evolution of socio-cognitive representations associated with sedentariness which will be essentially targeted by teachers (i.e., attitudes, norms and perceived behavioral control). One ambition of the CIPRES project is to create a motivational synergy in several contexts in which the child evolves towards the reduction of children's sedentary time.

In addition, according to a recent systematic review, interventions targeting the reduction of ST in children with a socio-ecological approach seem more effective when then involve several levels of actors. By involving a wide range of actors around children, the CIPRES study should target four levels of the SEM i.e., intrapersonal, interpersonal, organizational and community, which may be an important lever for its effectiveness, as shown in a recent systematic review [38]. In addition, according to this review [38], a combination of informational (e.g., key learning messages in curriculum) and behavioral strategies (e.g., setting screen-time goals) seems relevant, as planned in the CIPRES intervention.

We assume that this intervention will allow an average reduction of 30 minutes per day of the ST of children in the intervention vs control group. This intervention will aim to establish behaviors beneficial to health in children through community action involving multilevel actors of children's life. The choice to create an intervention by using a participatory approach will truly involve the key actors in the process of co-construction of the intervention and will sensitize a wide range of local actors to health promotion and, more particularly, sedentary prevention. Our intervention will have the originality of not being a top-down model (created by experts and applied in the field), but a collaborative model: the researchers propose relevant guidelines that have been proven in the literature, and the locals actors bring their experiential knowledge on the possible and appropriate forms of actions to adopt according to the context.

If the proposed intervention model provides conclusive evidence as to its co-construction, its implementation and its effectiveness in reducing sedentary lifestyle, it can then be deployed and transferred on a larger scale to reduce the sedentary lifestyle of French children of school age in the medium/long term.

The CIPRES project proposes the implementation of an original and innovative turnkey method to reduce ST among children year 4 and 5. The solid and innovative methodology integrates theoretical models that have been shown to act on the child's behavior through the intervention of several levels of influence, with the objective of modifying SB. If the results of this study show a significant impact on ST, this would provide new evidence and new avenues to explore in order to act effectively against SB. In addition, the co-construction was done to ensure good adaptability to the school program and context so that it could be deployed on a larger scale and over a longer period.

## Author Contributions

**Conceptualization:** Marion Carayol.

**Data curation:** Marion Carayol, Steven Laujac.

**Funding acquisition:** Marion Carayol.

**Investigation:** Marion Carayol, Steven Laujac, Marie Cholley-Gomez, Josselin Franceschi, Axel Rozand.

**Methodology:** Marion Carayol, Steven Laujac, Emma Guillet-Descas, Emmanuel Damville, Cyrille Delpierre.

**Project administration:** Marion Carayol, Laurence Pallier, Cyrille Delpierre.

**Resources:** Steven Laujac, Marie Cholley-Gomez, Josselin Franceschi, Axel Rozand, Jean-Pierre Estrella, Bruno Vanhierde.

**Supervision:** Marion Carayol, Laurence Pallier, Cyrille Delpierre.

**Writing – original draft:** Marion Carayol, Steven Laujac.

**Writing – review & editing:** Marion Carayol, Steven Laujac, Marie Cholley-Gomez, Josselin Franceschi, Axel Rozand, Laurence Pallier, Jean-Pierre Estrella, Bruno Vanhierde, Emma Guillet-Descas, Emmanuel Damville, Olivier Gavarry, Cyrille Delpierre.

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
