## [Decision Letter · Decision Letter 0]

6 Feb 2024

PONE-D-23-38897Co-construct, implement and evaluate a multi-level intervention to prevent a sedentary lifestyle in children - Study protocol of the CIPRES study.PLOS ONE

Dear Dr. laujac,

Thank you for submitting your manuscript to PLOS ONE. After careful consideration, we feel that it has merit but does not fully meet PLOS ONE’s publication criteria as it currently stands. Therefore, we invite you to submit a revised version of the manuscript that addresses the points raised during the review process.

We look forward to receiving your revised manuscript.

Kind regards,

Henri Tilga, PhD

Academic Editor

PLOS ONE

Journal Requirements:

Additional Editor Comments:

Overall, the Reviewer is satisfied with the manuscript. There are only some small comments from the Reviewer. Please see the Reviewer comments and revise the manuscript accordingly.

Reviewers' comments:

Reviewer's Responses to Questions

**Comments to the Author**

1. Does the manuscript provide a valid rationale for the proposed study, with clearly identified and justified research questions?

Reviewer #1: Yes

2. Is the protocol technically sound and planned in a manner that will lead to a meaningful outcome and allow testing the stated hypotheses?

Reviewer #1: Yes

3. Is the methodology feasible and described in sufficient detail to allow the work to be replicable?

Reviewer #1: Yes

4. Have the authors described where all data underlying the findings will be made available when the study is complete?

Reviewer #1: Yes

5. Is the manuscript presented in an intelligible fashion and written in standard English?

Reviewer #1: Yes

6. Review Comments to the Author

You may also provide optional suggestions and comments to authors that they might find helpful in planning their study.

Reviewer #1: This study protocol covers a cluster RCT of a comprehensive and complex intervention, which requires a disciplinary implementation and data collection. I have two minor comments:

- I checked the study populations in the intervention and control group and wondered how comparable these are as Brignoles is a little town in the rural area, whereas the other areas cover the more urban, coastal areas? The authors could elaborate a little bit more about potential and/or influential differences

- the co-construction of the intervention (line 317 -) is nice but I have little insight in the time frame used> did you start with involvement of parents then followed by teachers and later the community in a fixed time frame of was it at the same time in varying sequence of the stakeholders ? Please clarify

7. PLOS authors have the option to publish the peer review history of their article (what does this mean?). If published, this will include your full peer review and any attached files.

Reviewer #1: No

---

## [Author Response · Author response to Decision Letter 0]

7 Mar 2024

Response to reviewers:

This study protocol covers a cluster RCT of a comprehensive and complex intervention, which requires a disciplinary implementation and data collection. I have two minor comments:

- I checked the study populations in the intervention and control group and wondered how comparable these are as Brignoles is a little town in the rural area, whereas the other areas cover the more urban, coastal areas? The authors could elaborate a little bit more about potential and/or influential differences.

Thank you for your question. Indeed, the town of Brignoles is not a coastal town and has a rural population. The aim of the CIPRES project is to evaluate a sedentary prevention intervention in a wide sample including children from families with heterogeneous social and economic characteristics. To improve representativeness, we chose to characterise our recruitment by taking into account several parameters such as the ecologic deprivation index level (cf. Line 208) and the size of the town (population density). Thus, in order to favour a heterogeneous recruitment, more varied and representative of the Var, we chose to include rural areas (Brignoles), medium-sized towns (La Seyne-sur-Mer, Hyères) and a large town (Toulon) with different levels of EDI ranging from 1 to 5. We will statistically compare the levels of sedentariness in the different areas before the start of the implementation to evaluate if there are any differences according to area and EDI.

- the co-construction of the intervention (line 317 -) is nice but I have little insight in the time frame used> did you start with involvement of parents then followed by teachers and later the community in a fixed time frame of was it at the same time in varying sequence of the stakeholders ? Please clarify

Thank you for your question. As a first step, we identified “gate keepers” in order to make efficient contact with the targeted schools. We then organised meetings bringing together teachers, parents, children, educational advisers and educators in the same time frame. In this way, all the actors involved in the co-construction process were involved at the same time, meeting on a number of occasions. 

We added a sentence to clarify this point (cf. line 344-346, revised manuscript with track change): “All the actors were invited to take part in the co-construction at the same time frame, the meetings were spread over several sessions.”

---

## [Decision Letter · Decision Letter 1]

8 Apr 2024

Co-construct, implement and evaluate a multi-level intervention to prevent a sedentary lifestyle in children - Study protocol of the CIPRES study.

PONE-D-23-38897R1

Dear Dr. laujac,

We’re pleased to inform you that your manuscript has been judged scientifically suitable for publication and will be formally accepted for publication once it meets all outstanding technical requirements.

Kind regards,

Henri Tilga, PhD

Academic Editor

PLOS ONE

Additional Editor Comments (optional):

Reviewers' comments:

Reviewer's Responses to Questions

**Comments to the Author**

1. Does the manuscript provide a valid rationale for the proposed study, with clearly identified and justified research questions?

Reviewer #1: Yes

2. Is the protocol technically sound and planned in a manner that will lead to a meaningful outcome and allow testing the stated hypotheses?

Reviewer #1: Yes

3. Is the methodology feasible and described in sufficient detail to allow the work to be replicable?

Reviewer #1: Yes

4. Have the authors described where all data underlying the findings will be made available when the study is complete?

Reviewer #1: Yes

5. Is the manuscript presented in an intelligible fashion and written in standard English?

Reviewer #1: Yes

6. Review Comments to the Author

You may also provide optional suggestions and comments to authors that they might find helpful in planning their study.

Reviewer #1: I reread the paper with pleasure and I got an appropriate answer on my two questions. Therefore I advise to accept the paper.

7. PLOS authors have the option to publish the peer review history of their article (what does this mean?). If published, this will include your full peer review and any attached files.

Reviewer #1: No
